# Accuracy of computer-aided chest X-ray in community-based tuberculosis screening: Lessons from the 2016 Kenya National Tuberculosis Prevalence Survey

**Brenda Mungai** [1,2,3]*, **Jane Ong'angò**[4], **Chu Chang Ku**[5], **Marc Y. R. Henrion**[1,6], **Ben Morton**[1,7], **Elizabeth Joekes**[1,8], **Elizabeth Onyango**[9], **Richard Kiplimo**[9], **Dickson Kirathe**[9], **Enos Masini**[10,11], **Joseph Sitienei**[9], **Veronica Manduku**[4], **Beatrice Mugi**[12], **Stephen Bertel Squire**[1,13‡], **Peter MacPherson**[1,6,14‡], **The IMPALA Consortium**[¶]

1 Department of Clinical Sciences, Liverpool School of Tropical Medicine, Liverpool, United Kingdom, 2 African Institute for Development Policy, Nairobi, Kenya, 3 Centre for Health Solutions, Nairobi, Kenya, 4 Kenya Medical Research Institute (KEMRI), Nairobi, Kenya, 5 Department of Infectious Disease Epidemiology, Imperial College London, London, United Kingdom, 6 Malawi-Liverpool-Wellcome Trust Clinical Research Programme, Blantyre, Malawi, 7 Critical Care Department, Liverpool University Hospitals NHS Foundation Trust, Liverpool, United Kingdom, 8 Worldwide Radiology, Liverpool, United Kingdom, 9 Division of National Tuberculosis, Leprosy and Lung Disease Program, Nairobi, Kenya, 10 The Global Fund to Fight AIDS, Tuberculosis and Malaria, Geneva, Switzerland, 11 Stop TB Partnership, Le Grand-Saconnex, Switzerland, 12 Kenyatta National Hospital, Nairobi, Kenya, 13 Tropical & Infectious Diseases Unit, Liverpool University Hospitals NHS Foundation Trust, Liverpool, United Kingdom, 14 Clinical Research Department, London School of Hygiene and Tropical Medicine, Liverpool, United Kingdom

‡ These authors are joint senior authors on this work.
¶ Membership of the IMPALA Consortium is provided in the Acknowledgments section.
* brendanyambura2013@gmail.com

**Data Availability Statement:** The Kenya National Tuberculosis, Leprosy and Lung Disease program is the custodian of the 2016 Kenya Tuberculosis

## Abstract

Community-based screening for tuberculosis (TB) could improve detection but is resource intensive. We set out to evaluate the accuracy of computer-aided TB screening using digital chest X-ray (CXR) to determine if this approach met target product profiles (TPP) for community-based screening. CXR images from participants in the 2016 Kenya National TB Prevalence Survey were evaluated using CAD4TBv6 (Delft Imaging), giving a probabilistic score for pulmonary TB ranging from 0 (low probability) to 99 (high probability). We constructed a Bayesian latent class model to estimate the accuracy of CAD4TBv6 screening compared to bacteriologically-confirmed TB across CAD4TBv6 threshold cut-offs, incorporating data on Clinical Officer CXR interpretation, participant demographics (age, sex, TB symptoms, previous TB history), and sputum results. We compared model-estimated sensitivity and specificity of CAD4TBv6 to optimum and minimum TPPs. Of 63,050 prevalence survey participants, 61,848 (98%) had analysable CXR images, and 8,966 (14.5%) underwent sputum bacteriological testing; 298 had bacteriologically-confirmed pulmonary TB. Median CAD4TBv6 scores for participants with bacteriologically-confirmed TB were significantly higher (72, IQR: 58–82.75) compared to participants with bacteriologically-negative sputum results (49, IQR: 44–57, p<0.0001). CAD4TBv6 met the optimum TPP; with the threshold set to achieve a mean sensitivity of 95% (optimum TPP), specificity was 83.3%,

Prevalence Survey data. The prevalence survey data had been shared to PLOS One by the National TB program during the publication of the prevalence survey paper in the links below S1 Excel. Information line data for the Kenya TB prevalence survey 2016. https://doi.org/10.1371/journal.pone.0209098.s001 (CSV) S2 Excel. Data dictionary for the information line data for the Kenya TB prevalence survey 2016. https://doi.org/10.1371/journal.pone.0209098.s002 (XLS).

**Funding:** This research was funded by the National Institute for Health Research (NIHR) (IMPALA, grant reference 16/136/35) using UK aid from the UK Government to support global health research to SBS. SBS was Director of the IMPALA Global Health Research Unit at the time of this work and BNM was a PhD candidate funded through IMPALA. PM was funded by Wellcome (200901/Z/16/Z). MYRH was in part supported by a strategic award from Wellcome to Malawi-Liverpool-Wellcome Trust Clinical Research programme (206545/Z/17/Z). The views expressed in this publication are those of the author(s) and not necessarily those of the National Institute for Health Research or the UK Department of Health and Social Care. The funders had no role in study design, data collection and analysis, decision to publish, or preparation of the manuscript.

**Competing interests:** The authors have declared that no competing interests exist.

(95% credible interval [CrI]: 83.0%—83.7%, CAD4TBv6 threshold: 55). There was considerable variation in accuracy by participant characteristics, with older individuals and those with previous TB having lowest specificity. CAD4TBv6 met the optimal TPP for TB community screening. To optimise screening accuracy and efficiency of confirmatory sputum testing, we recommend that an adaptive approach to threshold setting is adopted based on participant characteristics.

## Introduction

With over 95% tuberculosis (TB) cases and deaths occurring in developing countries, there is need for substantially improved case detection to find the "missing millions" and accelerate action to achieve the sustainable development goals to end TB by 2030 [1–4]. Chest radiography (CXR) with computer-aided detection (CAD) software for TB has been recommended for systematic screening for tuberculosis disease in the 2021 WHO TB Screening Guidelines [5]. However, supporting data have predominantly come from clinical settings and not community-based settings, and CAD diagnostic accuracy is likely to vary considerably across different screening strategies and populations [6–8].

CXRs were used extensively in TB screening and active case finding (ACF) programmes in the mid-20th century due to high sensitivity (94%, 95% CI 88–98%), potential for high throughput, and lower infectious risk to health workers (compared to sputum collection for all) [9–12]. In addition, CXR can detect infectious but asymptomatic TB patients, this is important as a substantial fraction of TB transmission is attributable to the often prolonged asymptomatic infectious period [13]. Barriers to widespread CXR use include low specificity, limited access to high quality radiography equipment, critical shortage of radiologists in low- and middle-income countries (LMICs), and inter-and intra-observer variations during interpretation [11,12,14]. CAD software that provides a probabilistic score for TB offers a potential solution to these limitations [6,8,15]. Previous evaluations of CAD software have been mostly conducted in triage testing use situations, with very little data available to evaluate accuracy in community-based TB screening interventions [7,8,16].

Our aim was to evaluate the accuracy of the Computer-Aided Detection for Tuberculosis version 6 (CAD4TBv6) system for TB screening using a large data set (n = 61,848) from the 2016 Kenya National TB prevalence survey [17,18]. Our primary study outcome was to use a Bayesian modelling approach to evaluate the accuracy (sensitivity, specificity, and area under receiver-operator curve [AUC]) of CAD4TBv6 and Clinical Officer CXR interpretation against a modelled bacteriological reference standard. Our secondary study outcome was to estimate sensitivity and specificity for participants by age group, sex, cough of more than two weeks status and history of previous TB treatment, summarised as a function of CAD4TBv6 threshold. We hypothesized that CAD4TBv6 diagnostic sensitivity and specificity would meet the target product profile (TPP) for a test to identify people suspected of having TB, but that accuracy would vary between the general population compared to older age groups and people previously treated for TB, implying that an adaptive approach to CAD screening would be required [19].

## Methods

### Study design

We conducted a retrospective analysis of cross-sectional individual-level participant data from adult community members who participated in the 2016 Kenya National TB Prevalence

Survey [18]. The Kenya National TB prevalence survey followed WHO standard methods for prevalence surveys [20].

## Study population and Kenya TB prevalence survey procedures

The Prevalence Survey (reported elsewhere) [17,18] was a population-based national cross-sectional study conducted in 2015-2016.The target population for the survey comprised of all persons (male and female) aged 15 years and above, residing in Kenya and drawn from 100 clusters across the country. The aim of the survey was to determine the prevalence of bacteriologically confirmed pulmonary TB among adults aged 15 years and older and used the WHO-recommended screening strategy including symptom questionnaire and CXR [20]. HIV testing was not undertaken as part of the survey and participants self-reported their HIV status. There were 76,291 eligible participants out of whom, 63,050 (83%) were enrolled into the survey. Participation rates of females were higher than that of males at 87% and 77% respectively [17,18].

Prevalence survey participants completed a questionnaire to elicit TB symptoms. Subsequently, a digital posterior anterior CXR (CXDI, Delft Imaging, The Netherlands) was digitally acquired and uploaded to a digital archive. Study Clinical Officers who had received intensive one-week training in CXR interpretation independently read each film, blinded to each other's readings. In addition, one clinician was blinded to the participants' symptoms. (Clinical Officer characteristic are summarised in S1 Text). Clinical Officers classified CXRs as either: A) normal; B) abnormal, suggestive of TB; or C) abnormal other, in line with published definitions [18].

## Definitions

In the Prevalence Survey protocol, Clinical Officer field interpretations of CXRs undertaken were defined as: a) "normal" if there were no identifiable cardiothoracic or musculoskeletal abnormality; b) "abnormal, suggestive of TB" if there was any infiltrate or consolidation, non-calcified nodules, cavitary lesion, pleural effusion, hilar or mediastinal lymphadenopathy, linear or interstitial disease; or c) "Abnormal other" if there were features of any musculoskeletal abnormality, cardiac abnormality, any other pulmonary and pleural abnormality not consistent with TB, diaphragmatic, costophrenic angle blunting not related to effusion, solitary calcified nodules or lymph nodes [18]. In case of discordance between the two Clinical Officer readers on "abnormal, suggestive of TB," versus "abnormal other" or versus "normal," for the purpose of sputum eligibility the film was classified as "abnormal, suggestive of TB"; discordance of "normal" vs. "abnormal other" was classified as "abnormal other."

Participants with a CXR classified as "abnormal, suggestive of TB" by either one of the Clinical Officers, or with a cough of more than two weeks or who declined CXR screening were eligible for sputum collection as per the survey protocol [20]. A total of 9,715 (15%) participants were eligible for sputum collection. Participants were encouraged to produce sputum and were taken through some walking and breathing exercises. Two sputum samples (spot and following morning) were to be collected; out of the eligible participants, 9,120 (94%) had at least one sputum submitted.

The sputum samples collected were transported under cold chain to the National Tuberculosis Reference Laboratory. A direct smear was done on all samples with Auramine O stain followed by microscopic examination using a fluorescent microscope. Xpert MTB/Rif testing and solid culture using Löwenstein-Jensen medium (incubation at 37˚C) were conducted on all samples as per the survey procedures. Culture was reported as negative if there was no growth after eight weeks. To confirm the presence of *Mycobacterium tuberculosis* complex, all visible

colonies grown on culture media were confirmed by acid-fast bacilli (AFB) microscopy and tested for presence of *Mycobacterium* protein 64 (MPT64) by SD Bio line Immunochromatographical assay. Geno-Type *Mycobacterium* AS (Hain Life Science) test kits were used to identify presence of non-tuberculous Mycobacteria [17,18].

The case definition of a pulmonary TB case was a participant who was eligible for the survey and had sputum bacteriologically confirmed for MTB (Xpert and/or culture positive). Though smear microscopy was conducted for all collected sputum, this was not included in the case definition. Participants with bacteriologically-confirmed pulmonary TB (Xpert and/or culture-positive) were referred for TB treatment; participants with CXR abnormalities and no bacteriological evidence of TB were referred to health facilities for clinical assessment. As HIV testing was not undertaken as part of TB prevalence survey activities, in line with the national guidelines, participants referred for TB treatment were offered HIV testing at referral facilities. The survey found a weighted national pulmonary TB prevalence of 558 (95% CI 455–662) per 100,000 adult population [17,18].

## Study procedures and definitions

Analysis was conducted between January 2020 and October 2021. Anonymised, compressed DICOM CXR images were uploaded from the prevalence survey digital archive to the Delft Imaging CAD4TB cloud server and analysed using CAD4TBv6 [21]. Results were provided as probabilistic scores, ranging from 0 to 99, with higher scores closer to 100 indicating a greater probability of TB. The reference standard for this analysis was bacteriologically-confirmed TB, defined as Xpert and/or culture positive with MTB speciation. Analysis was conducted independently; the commercial provider (Delft Imaging) was not part of the study team and had no role in study design, data collection, analysis, or interpretation of results.

## Statistical analysis

The characteristics of prevalence survey participants were summarized using means (with standard deviations), medians (with interquartile ranges), and percentages, and compared by Clinical Officer CXR interpretation. We used the Kruskall-Wallis test to investigate differences in CAD4TBv6 scores between Clinical Officer interpretation groups, and Chi-square and Fisher's exact tests for categorical participant characteristics. Distributions of CAD4TBv6 scores were summarized by medians and 95% highest density intervals (HDI) and compared by whether sputum was collected or not, and by sputum bacteriological status.

For our primary study outcome, we compared the accuracy (sensitivity, specificity, and area under receiver-operator curve [AUC]) of CAD4TBv6 with a modelled bacteriological reference standard. As collection of sputum was conditional on either a participant reporting having cough of two weeks or greater or a Clinical Officer CXR classification of "abnormal, suggestive of TB", Bayesian latent class modelling was employed to infer disease prevalence within the portion of the study population without TB symptoms or CXR signs suggestive of TB, and to estimate the sensitivity, specificity, and AUC of CAD4TBv6 at thresholds ranging from 0 to 99. The model also outputs estimates of the underlying status of active pulmonary TB, and the sensitivity and specificity of Clinical Officer CXR interpretation as a screening tool and of sputum bacteriological results for the underlying true TB status. Full model details and diagnostics are reported in S2 Text. We placed informative priors on the overall prevalence of TB, inferred by the prevalence survey results, and weakly informative priors on other model parameters, including presence of cough of greater than two weeks, age, sex, and history of previous TB treatment. To aid model convergence, we fixed specificity of the combined bacteriological reference standard (Xpert and/or culture positive) to be 99%.

Models were fitted in Stan using the cmdstanr interface [22], with convergence assessed by inspecting trace plots across three sampling chains and Gelman-Rubin statistics. Inference was based on summarising 12,000 post-warm-up samples. We plotted model posterior summary estimates of sensitivity and specificity across CAD4TBv6 thresholds, and compared to optimum (sensitivity: 95%, specificity: 80%) and minimum (sensitivity: 90%, specificity: 70%) TPP for a community or referral test to identify people suspected of having TB [19]. In secondary analysis, we restricted model sensitivity and specificity estimates for participants by age group, sex, cough of more than two weeks status and history of previous TB treatment, and summarised as a function of CAD4TBv6 threshold, estimating what accuracy would be achieved by setting an overall screening CAD4TBv6 threshold to achieve the optimum TPP sensitivity cut-off (95%). We did not stratify by HIV status, as there was substantial missing data and testing was not performed in the prevalence survey. All analysis was done using R v4.1.1 (R Foundation for Statistical Computing, Vienna).

## Ethical considerations

This study was conducted as part of the Kenya Prevalence survey ethics approval reference number SSC 2094 by the Kenya Medical Research Institute. Prevalence survey participants provided written informed consent for survey activities. Additional approval was obtained from the Division of National Tuberculosis, Leprosy and Lung Disease program which is the custodian of the TB prevalence survey data. The data was processed within the Kenya Health Informatics System Governance [23] and General Data Protection Regulations [24]. All CXR images were deleted from the Delft Imaging server after analysis as contractually stipulated.

## Results

### Participant characteristics and clinical officer chest X-ray interpretation

A total of 62,484 CXR images were uploaded for CAD4TBv6 processing. After exclusion of 636 images that were either not analysable by the CAD4TBv6 software or had missing clinical data, 61,848 (99.0%) were included for analysis (Fig 1).

Of the 61,848 participants whose images were analysed, 58.5% (36,187) were women and 70.7% (43,754) were aged <45 years (Table 1). Two thousand and eighty-four (3.4%) had previously been treated for TB, and 58 (0.1%) were currently being treated for TB. Overall, HIV positive status was self-reported by 1,577/31,495 (5.0%) of participants with data available.

Clinical Officers classified 50,045 (80.9%) CXRs as "normal", 5,045 (8.2%) as "abnormal, other", and 6,758 (10.9%) as "suggestive of TB" (Table 1). Compared to participants with CXRs classified by Clinical Officers as "normal" or "abnormal, other," participants with CXRs classified as "suggestive of TB" were more likely to be men (50.5%), and have been previously treated for TB (50.3%). CAD4TBv6 scores were significantly higher among participants whose CXR were classified as "suggestive of TB" (median: 52, IQR: 46–62), compared to those classified as "normal" (median: 43, IQR: 24–46) or "abnormal, other" (median: 48, IQR: 45–53), p<0.0001.

### CAD4TBv6 scores by sputum testing and bacteriological TB status

Sputum was collected from 8,996/61,848 (14.5%) participants, of whom 298 (3.3%) had bacteriologically-confirmed TB. CAD4TBv6 scores were higher among participants who had sputum tested in the prevalence survey (median: 49, 95%HDI: 9–82) than where sputum was not tested (median: 44, 95%HDI: 4–55)—Fig 2. The median CAD4TBv6 score for participants with bacteriologically -confirmed TB was substantially higher at 72 (95%HDI: 38–98) compared to 49 (95%HDI: 8–79) for participants with bacteriologically-negative sputum results.

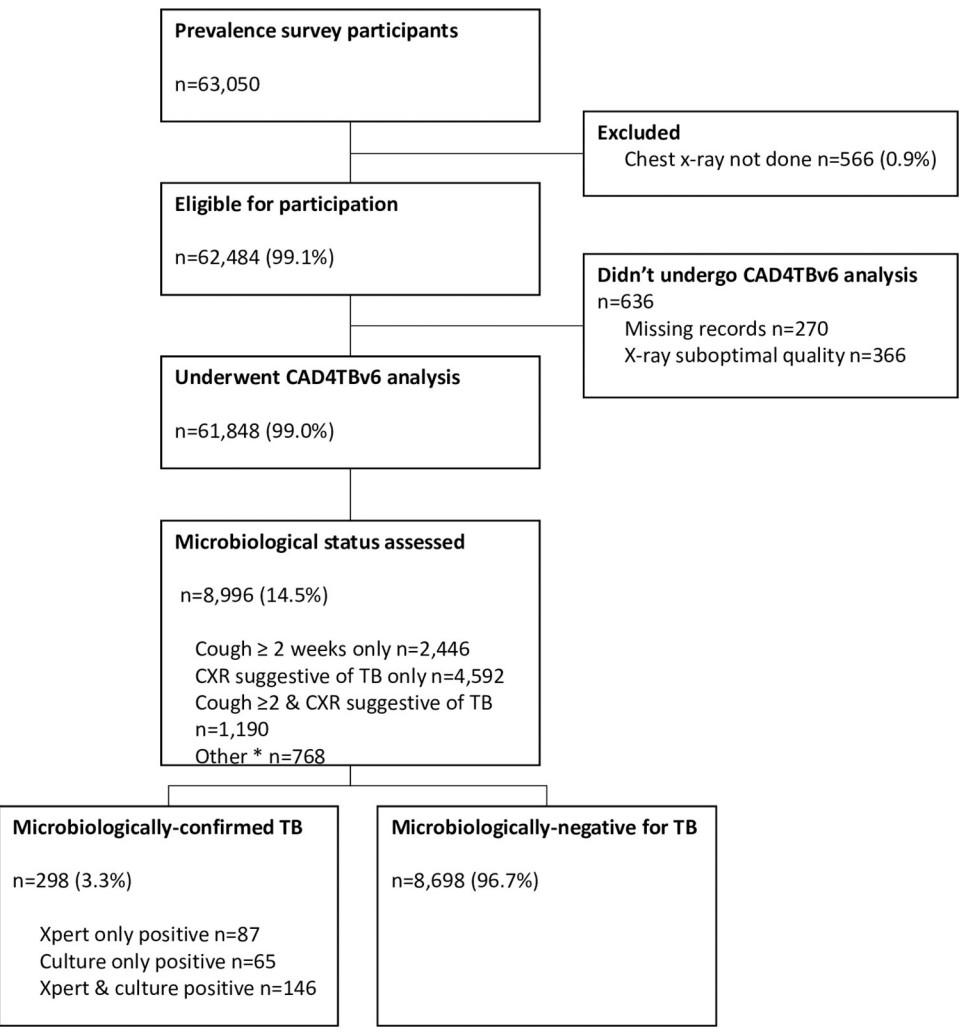

**Fig 1. Participant flowchart and results of prevalence survey investigations.**

Overall, 4,678/8,996 (52.0%) of participants with sputum collected were women and the median age was 45 years (IQR: 30–61). Among those tested, more men had bacteriologically-confirmed TB (185/4318, 4.3%) than women (113/4678, 2.4%, p<0.0001). Conditional on being tested, bacteriologically-confirmed TB was higher among those with cough of more than two weeks (3.9%, 140/3636) compared to those with no cough or cough of less than two weeks (2.9%, 158/5360, p = 0.022), and among participants with weight loss (5.8%, 39/671) compared to those without weight loss (3.1%, 259/8325, p<0.0003). Conditional on testing, participants who had previous TB treatment were more likely to be bacteriologically positive (7.3%, 81/1109) than those who had not been previously treatment (2.8%, 217/7887, p<0.0001).

## Model estimated TB prevalence and accuracy of clinical officer CXR interpretation, sputum testing and CAD4TBv6 screening

Model Gelman-Rubin statistics were all <1.01 and trace plots showed good mixing across chains (S2 Text). Mean posterior estimates of the prevalence of bacteriologically-confirmed pulmonary TB were 568 per 100,000 (95% credible interval [CrI]: 478–654) overall, 927 per 100,000 (95% CrI: 783–1090) for men, and 314 per 100,000 (95% CrI: 246–391) for women.

**Table 1. Characteristics of participants in Kenya National TB prevalence survey, by clinical officer chest X-ray interpretation.**

| | Normal (N = 50,045) | Abnormal, other (N = 5,045) | Suggestive of TB (N = 6,758) | Total (N = 61,848) | P-value[†] |
|---|---|---|---|---|---|
| **Age group** | | | | | < 0.001 |
| 15–24 years | 16,305 (32.6%) | 403 (8.0%) | 742 (11.0%) | 17,450 (28.2%) | |
| 25–34 years | 13,400 (26.8%) | 627 (12.4%) | 1,167 (17.3%) | 15,194 (24.6%) | |
| 35–44 years | 9,072 (18.1%) | 765 (15.2%) | 1,273 (18.8%) | 11,110 (18.0%) | |
| 45–54 years | 5,553 (11.1%) | 831 (16.5%) | 1,032 (15.3%) | 7,416 (12.0%) | |
| 55–64 years | 3,210 (6.4%) | 901 (17.9%) | 957 (14.2%) | 5,068 (8.2%) | |
| 65+ years | 2,505 (5.0%) | 1,518 (30.1%) | 1,587 (23.5%) | 5,610 (9.1%) | |
| **Sex** | | | | | < 0.001 |
| Women | 29,432 (58.8%) | 3,410 (67.6%) | 3,345 (49.5%) | 36,187 (58.5%) | |
| Men | 20,613 (41.2%) | 1,635 (32.4%) | 3,413 (50.5%) | 25,661 (41.5%) | |
| **HIV-status**[*] | | | | | |
| Missing | 23,895 | 2871 | 3,587 | 30,353 | |
| HIV-negative | 24,985 (95.5%) | 2,055 (94.5%) | 2,878 (90.8%) | 29,918 (95.0%) | < 0.001 |
| HIV-positive | 1,165 (4.5%) | 119 (5.5%) | 293 (9.2%) | 1,577 (5.0%) | |
| **Cough** | | | | | < 0.001 |
| No | 43,989 (87.9%) | 4,065 (80.6%) | 4,752 (70.3%) | 52,806 (85.4%) | |
| Yes | 6,056 (12.1%) | 980 (19.4%) | 2,006 (29.7%) | 9,042 (14.6%) | |
| **Cough>2 weeks** | | | | | < 0.001 |
| No | 47,859 (95.6%) | 4,504 (89.3%) | 5,484 (81.1%) | 57,847 (93.5%) | |
| Yes | 2,186 (4.4%) | 541 (10.7%) | 1,274 (18.9%) | 4,001 (6.5%) | |
| **Fever** | | | | | < 0.001 |
| No | 47,088 (94.1%) | 4,408 (87.4%) | 5,566 (82.4%) | 57,062 (92.3%) | |
| Yes | 2,957 (5.9%) | 637 (12.6%) | 1,192 (17.6%) | 4,786 (7.7%) | |
| **Weight loss** | | | | | < 0.001 |
| No | 49,173 (98.3%) | 4,921 (97.5%) | 6,212 (91.9%) | 60,306 (97.5%) | |
| Yes | 872 (1.7%) | 124 (2.5%) | 546 (8.1%) | 1,542 (2.5%) | |
| **Night sweats** | | | | | < 0.001 |
| No | 45,546 (91.0%) | 4,046 (80.2%) | 5,077 (75.1%) | 54,669 (88.4%) | |
| Yes | 4499 (9.0%) | 999 (19.8%) | 1,681 (24.9%) | 7,179 (11.6%) | |
| **Current TB treatment** | | | | | < 0.001 |
| No | 50,023 (100.0%) | 5,043 (100.0%) | 6,724 (99.5%) | 61,790 (99.9%) | |
| Yes | 22 (0.0%) | 2 (0.0%) | 34 (0.5%) | 58 (0.1%) | |
| **Previous TB treatment** | | | | | < 0.001 |
| No | 49,171 (98.3%) | 4,883 (96.8%) | 5,710 (84.5%) | 59,764 (96.6%) | |
| Yes | 874 (1.7%) | 162 (3.2%) | 1,048 (15.5%) | 2,084 (3.4%) | |
| **CAD4TBv6 score** | | | | | < 0.001 |
| Count | 50,045 | 5,045 | 6,758 | 61,848 | |
| Median | 43.0 | 48.0 | 52.0 | 44.0 | |
| Q1, Q3 | 24.0, 46.0 | 45.0, 53.0 | 46.0, 62.0 | 27.0, 48.0 | |

[*]HIV testing was not undertaken in prevalence survey. Number show participants who had their self-reported HIV status collected.

[†]P-values calculated by Kruskull-Wallis test for CAD4TBv6 scores, Fisher's exact test for current TB treatment, and Chi-square test for all other categorical variables.

From the model, we estimated that the overall sensitivity of Clinical Officer CXR interpretation as "suggestive of TB" for bacteriologically-confirmed TB was 43.7% (95% CrI: 23.8–66.4%) and specificity was 89.2% (89.0%-89.6%)–Table 2. However, in stratified analysis, sensitivity of Clinical Officer CXR interpretation was considerably higher (84.8%, 95% CrI: 70.0–

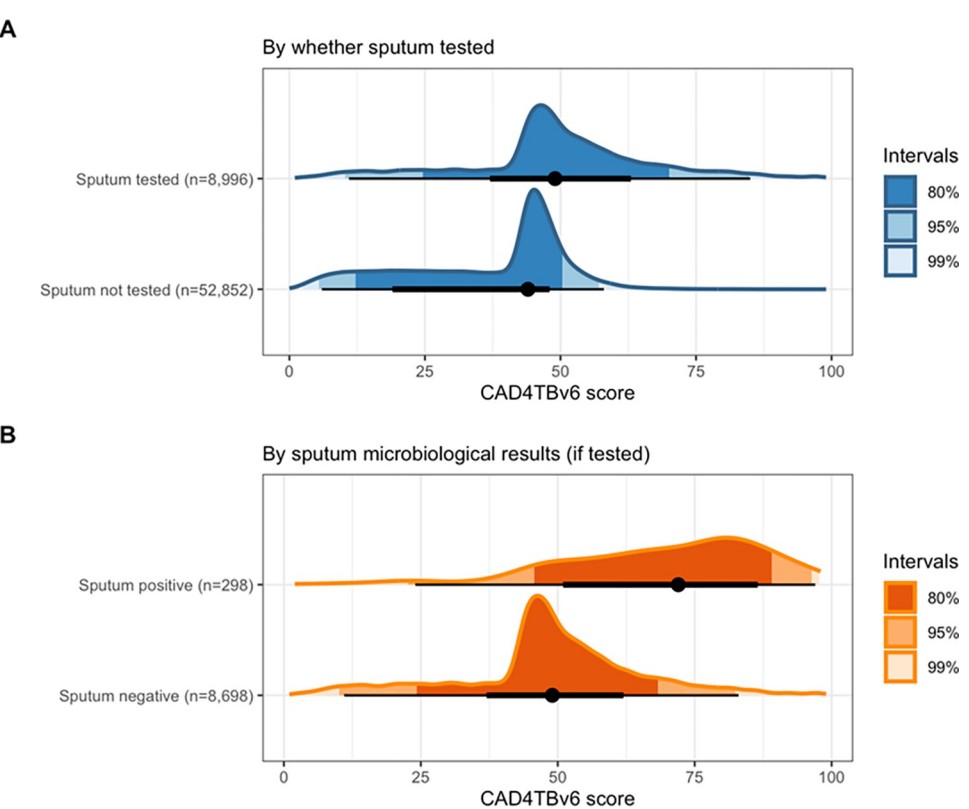

**Fig 2. Distribution of CAD4TBv6 scores in Kenya National TB prevalence survey.** A) Distribution (median and 95% highest density interval) of CAD4TBv6 scores by whether prevalence survey participant's sputum was tested or not. B) Distribution (median and 95% highest density interval) of CAD4TBv6 scores by sputum bacteriological results. 95%HDI: 95% highest density interval.

94.2%) and specificity was lower (54.1% 95% CrI: 51.3–56.9%) among participants who had previously been treated for TB. Accuracy was similar when stratified by sex and cough of more than two weeks status. The posterior mean sensitivity of the modelled combined sputum or Xpert or culture reference standard for the true underlying TB status was estimated to be 70.0% (95% CrI: 56.8%-84.6%) overall, and was 97.0% (95% CrI: 96.5%-97.5%) among participants with cough for two weeks or longer.

The posterior mean estimate of the AUC for CAD4TBv6 was 96.7% (95% CrI: 95.9%-97.4%). With CAD4TBv6 thresholds set to achieve a sensitivity of 90% (minimum TPP) or 95% (optimum TPP) mean specificity was 90.4% (95% CrI: 88.1%-92.6%) and 83.2% (95% CrI: 79.9%-86.6%) respectively–Fig 3. With CAD4TBv6 thresholds set to achieve a specificity of 70% (minimum TPP) or 80% (optimum TPP), mean sensitivity was 98.3% (95% CrI: 97.6–99.8%) and 96.2% (95.0–97.3%).

When model estimates were stratified by participant characteristics (age, sex, presence of cough of more than two weeks, history of previous TB), we found substantial variation in the sensitivity and specificity of CAD4TBv6 for modelled bacteriologically-confirmed pulmonary TB (Fig 4). With the CAD4TBv6 threshold set to 55 to achieve overall sensitivity of 95% for the optimal TPP within the prevalence survey population, sensitivity was highest among participants aged 41 years and older, who had previously been treated for TB and who had cough for more than two weeks. In contrast, specificity was lowest among participants previously treated for TB, and among older participants.

**Table 2. Model-based accuracy of screening and diagnostic tests per category compared to the modelled true status.**

|  | Sensitivity (%) | 95% CrI | Specificity (%) | 95% CrI |
|---|---|---|---|---|
| **Clinical Officer CXR interpretation "Suggestive of TB"** |  |  |  |  |
| Overall | 43.7% | 23.8–66.4% | 89.2% | 89.0–89.6% |
| Men | 48.4% | 27.7–71.0% | 87.1% | 86.6–87.5% |
| Women | 40.3% | 21.0–63.2% | 90.9% | 90.6–91.2% |
| Cough >2weeks | 53.5% | 33.1–74.2% | 83.1% | 82.7–83.6% |
| No cough >2weeks | 43.0% | 23.1–65.9% | 89.7% | 89.5–90.0% |
| Previous TB | 84.8% | 70.0–94.2% | 54.1% | 51.3–56.9% |
| No previous TB | 42.2% | 22.1–65.4% | 90.5% | 90.3–90.8% |
| **Sputum Xpert or culture** |  |  |  |  |
| No cough >2weeks | 70.0% | 56.8–84.6% | 99% (fixed) | – |
| Cough >2weeks | 97.0% | 96.5–97.5% | 99% (fixed) | – |
| **Against minimum Target Product Profile** |  |  |  |  |
| CAD4TBv6 (for sensitivity 90%) CAD: 61 | 90% (fixed) | – | 90.4% | 88.1–92.6% |
| CAD4TBv6 (for specificity 70%) CAD: 47 | 98.3% | 97.6–98.8% | 70% (fixed) | – |
| **Against optimum Target Product Profile** |  |  |  |  |
| CAD4TBv6 (for sensitivity 95%) CAD: 55 | 95% (fixed) | – | 83.2% | 79.9–86.6% |
| CAD4TBv6 (for specificity 80%) CAD: 53 | 96.2% | 95.0–97.3% | 80% (fixed) | – |

CrI: Credible interval.

## Discussion

This is the first study, to the best of our knowledge, to evaluate the accuracy of computer-aided CXR screening for TB in a community-based prevalence survey. Highly specific Xpert and culture tests were used as the bacteriological reference standard [25], with Bayesian latent class modelling employed to infer disease prevalence within the portion of the study population without TB symptoms or CXR signs suggestive of TB. Overall in the screening population, CAD4TBv6 met both the minimum and optimum TPP for a community-based referral test for identifying people suspected of having TB [19]. Very high sensitivity of 98–100% was demonstrated in participants in older age groups (41 years or older), those with reported cough>2 weeks and participants with previous TB history. Conversely, participants in older age groups and those with previous TB history had lower specificity ranging from approximately 38–70%. Computer-aided CXR screening is an accurate tool that could be used to support community TB screening in high burden countries where access to radiologists and clinicians is limited. To optimise screening accuracy and efficiency of confirmatory sputum testing, we recommend that an adaptive approach to screening threshold definition is adopted based on participant characteristics.

Community-based active case finding (ACF) for TB is effective at reducing TB prevalence if delivered with sufficient and sustained intensity to high burden populations [25,26]. However, operationalisation of ACF in a resource limited setting has been challenging due to substantial resourcing requirements and suboptimal TB screening and diagnosis tests [12,27]. The availability of portable/ultra-portable CXRs and CAD offer a potential solution to conduct community-based ACF for at risk groups in densely populated urban areas where TB transmission is now concentrated [8]. We have demonstrated that, overall within the prevalence survey population, CXR based screening in combination with CAD is accurate. However, due to low specificity in older age groups and persons with previous TB history, and adaptive CAD threshold could be more efficient than current approach of a single threshold for all participants. CAD

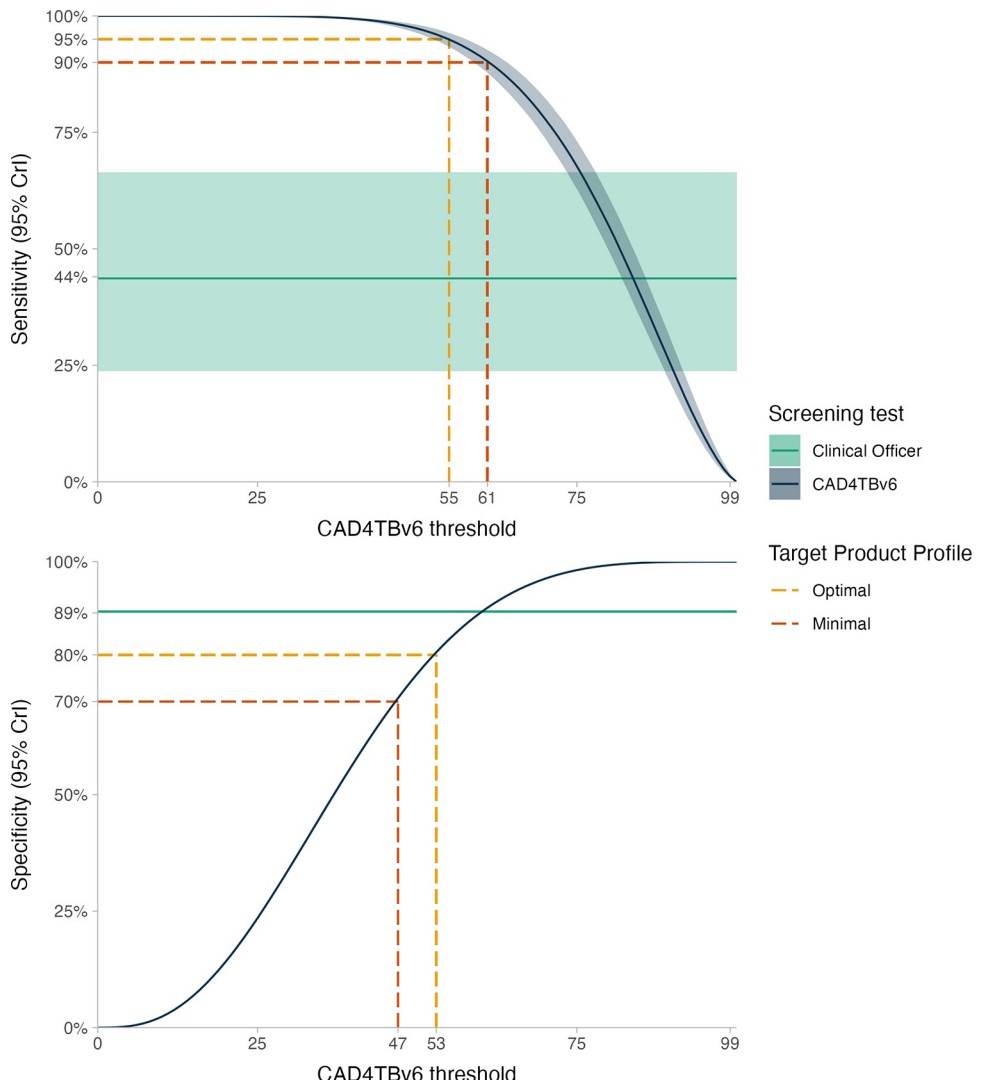

**Fig 3. Model-based sensitivity and specificity of CAD4TBv6 for bacteriologically-confirmed pulmonary TB at minimum and optimum target product profile thresholds.**

gives the additional flexibility for TB programs to vary the threshold for sputum testing [8]. Given the limited resources available to National TB Programmes, by varying the CAD screening threshold, the number of TB cases deemed acceptable to be missed can be balanced against how much money is available to spend on expensive confirmatory sputum investigation [8]. By adopting an adaptive threshold within population groups, we believe that further gains in accuracy and programme efficiency can be gained. CXR and CAD as tools for community-based TB screening ACF, additionally offer the potential for individuals and TB programmes, including: earlier diagnosis; identification of asymptomatic TB, potentially reducing transmission; reductions in false positive bacteriological tests with harm from prolonged unnecessary treatment; and reduction in catastrophic costs [8,13,28].

The prevalence survey participants are representative of the general population as they were randomly selected with a high participation rate, though higher amongst women than men [17,20]. The higher participation rate in women is a finding similar to other TB prevalence surveys [29]. The CAD accuracy finding in our study is therefore likely to be generalizable to

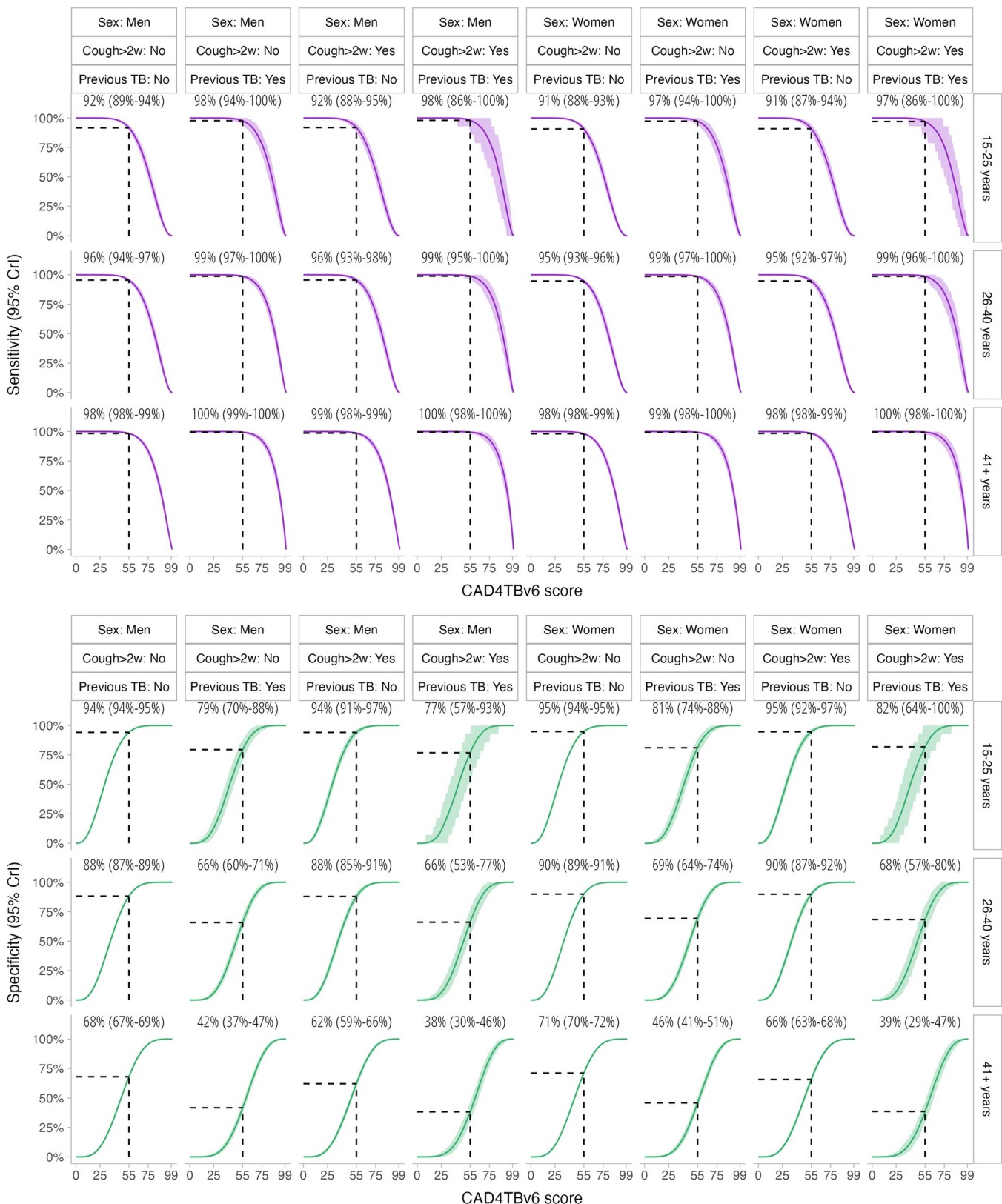

**Fig 4. Sensitivity and specificity of CAD4TBv6 by prevalence survey participant characteristics, with threshold set at optimal target product profile to achieve overall sensitivity of 95% (CAD4TBv6 = 55).**

countries in sub-Saharan Africa with high burden of TB and HIV. Though our study focused on one software (CAD4TBv.6), other comparable software that had the CE (Conformité Européenne) marking by January 2020 (Lunit Insight CXR, Lunit Insight; and qXR v2, Qure.ai.) may perform similarly or better than this [5,8]. Rapid advances have been made in CAD software development, with a total of 12 software solutions identified in March 2021 and version updates occurring frequently [8,30]. For example, this analysis was done with a version of CAD4TBv6 software which is no longer commercially available. This has been replaced by a newer version CAD4TB v7 with reported enhanced analysis [31]. Regular updating of WHO guidelines is therefore required to keep pace with these advances. As national TB programs adopt CAD technology into screening activities, in addition to performance, other implementation considerations include; cost effectiveness, compatibility of the X-ray systems, input image format, connectivity requirement and offline operation option, integration with any patient archiving systems, customer service and support, data protection, and ability to detect other non TB conditions [8,30,32]. Conditions other than TB may be as, or more prevalent than TB in high TB prevalence settings, and require comprehensive approaches to ensure participants in TB screening programmes are linked to appropriate care [33]. TB screening programs should plan for this and take into consideration resource implications to ensure additional health benefits through the identification of populations at risk of diseases other than TB. In addition to diagnostic accuracy; clinical utility, acceptability and feasibility of using CAD should be assessed [34].

In our secondary analysis we found that accuracy varied considerably by participant characteristics, specifically age and previous TB history. Similar to a previous study in Bangladesh among adults attending primary health care for triage setting, there was no significant difference in performance of CAD4TBv6 between men and women [8]. The lower specificity of CAD4TBv6 in the older age groups and those with prior history of TB is a finding similar to previous studies [8,16]. There are numerous anatomical and pathophysiological changes occurring in old age that could explain this lower age-related specificity, including age-related changes and sequalae of life-course accumulated lung damage [35]. People with prior TB have lung changes that could lead to difficulty distinguishing old vs active disease, leading to low specificity [8,16]. Further algorithm training with images from older populations may result in refinement of CAD software with improvements in specificity. In addition, two stage screening of CAD with symptom screen followed by C-reactive protein or other novel screening tests in older populations and in participants with previous TB history could improve specificity, although this requires further investigation.

In prevalence surveys, image classification criteria are set to a low threshold for referral for sputum testing, and non-expert readers like Clinical Officers are trained to interpret with higher sensitivity and lower specificity to avoid missing prevalent TB cases [20]. We found that, overall, the sensitivity of Clinical Officer CXR interpretation ("suggestive of TB") for the underlying true TB status was lower (44%) and specificity was higher (89%) than anticipated [5], but that sensitivity was substantially higher (84%) and specificity lower (54%) among participants with previous TB, and with no appreciable differences by other participant characteristics. This overall low sensitivity is not usually identified by other analyses that compare clinical CXR interpretation to a microbiological reference standard, and that assume that sputum testing is 100% sensitive. From our latent class model, we can then infer that true TB cases that are bacteriologically-negative are likely to have minimal or no CXR abnormalities (unless previously treated for TB), and so are currently undetectable without new, more sensitive TB diagnostic tests. As in other studies, we have demonstrated that CAD at varied thresholds achieves higher sensitivity than human readers [6,8,14]. CAD has a high throughput and has been shown to reduce the time to treatment [28,32]. Additionally, CAD has the benefit of

flexibility of varying thresholds, with a higher threshold improving the positive predictive value and reducing the number of Xpert tests required to diagnose a patient by up to 50% while maintaining sensitivity above 90% [8]. For TB prevalence surveys, we recommend that based on accuracy, a strategy including CAD should be considered, supported by formal health economic analyses to determine the health system feasibility of wide scale implementation. Mathematical modelling studies will likely be required to investigate potential impact on longer term trends of TB incidence, prevalence and mortality.

A major strength of our study is the use of a large population based data set from a well-conducted, WHO-approved TB prevalence survey [17,20]. Analysis of the CXRs was blinded to bacteriological status, and we used a robust bacteriological reference standard [36]. In addition, we used latent class model to estimate accuracy, recognising that completion of the reference standard (microbiological testing for pulmonary TB) was conditional on participants having an abnormal chest X-ray suggestive of TB or TB symptoms. Though TB prevalence surveys are designed to be as accurate as possible, they are limited by cost. Extremely large prevalence surveys needed to provide reasonable precision (usually +/-10%, as per the WHO standards for prevalence survey) [20] are rarely affordable. Sputum microbiological testing is the major driver of cost in prevalence surveys, and so programmes must significantly limit the numbers of sputum tests done to make it feasible. This is done by having a two-stage screening approach, recognising that some prevalent TB may be missed. In our analysis we were able to utilise all data available for all the ~62,000 participants in the prevalence survey, rather than just the ~8000 who underwent a sputum microbiological test. In a programmatic prevalence survey or community-based active case finding intervention, CAD would be used as an initial screening test among all participants. As prevalence survey participants who underwent sputum testing have substantially different characteristics and prevalence of disease from participants who did not, non-inclusion of these participants is likely to bias estimates of accuracy. Adopting the latent class modelling approach therefore provides improved estimates of accuracy of CAD for the target population (community members participating in TB screening programmes), rather than focusing on only those whose sputum was tested. Our model prevalence estimates are slightly higher than empirical estimates obtained from the prevalence survey (especially for men: 927 per 100,000 [783–1090] in our analysis vs. 809 per 100,000 [656–962] in the weighted prevalence survey estimate); (17) this is to be anticipated as the primary prevalence survey did not undertake microbiological testing of all participants. Limitations include our study participants being aged 15-years and above, therefore we cannot comment on the diagnostic accuracy of CAD4TBv6 in children. Additionally, the study only included bacteriologically-confirmed TB; we did not undertake the assessment of accuracy in participants with sputum negative TB (clinically diagnosed). We also were not able to stratify performance by HIV status as testing was not systematically conducted during the prevalence survey. This would have been important for an in-depth subgroup analysis in a high TB-HIV prevalence setting. Kenya has a HIV prevalence of 4.9% with approximately 1.6 million people living with HIV and an estimated HIV-positive TB incidence at 70/100,000 [1,37]. We therefore expected a lower CAD specificity in our setting as CXR is known to be less sensitive in immunocompromised patients with pulmonary TB [38–41]. We recommend further evaluation of CAD software in high TB-HIV prevalence settings and further studies on accuracy within HIV-positive populations.

In conclusion, the END TB strategy calls for concerted efforts to improve diagnosis of TB, including through new and effective approaches to systematic screening. We have demonstrated that CAD4TBv6 is an accurate tool for community based TB screening in Kenya and met the TPP in this population. Our findings support the 2021 WHO recommendations on use of CAD in place of human readers, in resource limited settings where radiologists are

scarce, for interpreting digital chest X-rays for screening and triage for TB disease among individuals aged 15 years and older in populations in which TB screening is recommended [5]. However, we recommend an adaptive approach to setting screening thresholds as this could further improve screening accuracy and efficiency.

## Supporting information

**S1 Text. Prevalence Survey Clinical officers' qualifications and work experience.**
(PDF)

**S2 Text. Modelling CAD4TBv6 accuracy.**
(HTML)

## Acknowledgments

We are grateful to the 2016 Kenya Tuberculosis Prevalence Survey team and the Division of National Tuberculosis, Leprosy and Lung Disease Program whose data we used for this secondary study. We would like to thank Martin Githiomi- Information Technology specialist who helped in review of the DELFT-NTLD-AFIDEP contracts ensuring the data protection act was adhered to. We also acknowledge Wendy Nkirote who played a role in review of the methodology part of the prevalence survey laboratory processes to ensure it was captured accurately. Finally Maureen Kamene (Former Head NTLD Program), Aiban Ronoh (Head of department, Monitoring and Evaluation) and Jeremiah Okari (Head of department- Diagnostics) who facilitated the process of accessing the prevalence survey data.

The IMPALA Consortium membership: Emmanuel Addo-Yobo, Brian Allwood, Hastings Banda, Imelda Bates, Amsalu Binegdie, Muhwa Jeremiah Chakaya, Asma El Sony, Adegoke Falade, Jahangir Khan, Maia Lesosky, Bertrand Mbatchou, Hellen Meme, Kevin Mortimer, Beatrice Mutayoba, Louis Niessen, Nyanda Elias Ntinginya, Angela Obasi, Jamie Rylance, Miriam Taegtmeyer, Rachel Tolhurst, William Worodria, Heather Zar, Eliya Zulu, Lindsay Zurba

## Author Contributions

**Conceptualization:** Brenda Mungai, Jane Ong'angò, Ben Morton, Elizabeth Joekes, Enos Masini, Stephen Bertel Squire, Peter MacPherson.

**Data curation:** Brenda Mungai, Richard Kiplimo, Dickson Kirathe, Peter MacPherson.

**Formal analysis:** Chu Chang Ku, Marc Y. R. Henrion, Peter MacPherson.

**Funding acquisition:** Stephen Bertel Squire.

**Investigation:** Brenda Mungai, Elizabeth Joekes, Richard Kiplimo, Dickson Kirathe, Joseph Sitienei, Stephen Bertel Squire, Peter MacPherson.

**Methodology:** Brenda Mungai, Jane Ong'angò, Ben Morton, Elizabeth Joekes, Richard Kiplimo, Dickson Kirathe, Enos Masini, Joseph Sitienei, Veronica Manduku, Beatrice Mugi, Stephen Bertel Squire, Peter MacPherson.

**Project administration:** Brenda Mungai, Elizabeth Onyango, Dickson Kirathe.

**Resources:** Stephen Bertel Squire.

**Supervision:** Jane Ong'angò, Ben Morton, Veronica Manduku, Stephen Bertel Squire.

**Validation:** Brenda Mungai, Stephen Bertel Squire, Peter MacPherson.

**Visualization:** Brenda Mungai, Chu Chang Ku, Marc Y. R. Henrion, Peter MacPherson.

**Writing – original draft:** Brenda Mungai, Ben Morton, Elizabeth Joekes, Stephen Bertel Squire, Peter MacPherson.

**Writing – review & editing:** Brenda Mungai, Jane Ong'angò, Chu Chang Ku, Marc Y. R. Henrion, Ben Morton, Elizabeth Joekes, Elizabeth Onyango, Richard Kiplimo, Dickson Kirathe, Enos Masini, Joseph Sitienei, Veronica Manduku, Beatrice Mugi, Stephen Bertel Squire, Peter MacPherson.

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
