## [Decision Letter · Decision Letter 0]

28 Feb 2022

PGPH-D-22-00030

Accuracy of computer-aided chest X-ray screening in the Kenya National Tuberculosis Prevalence Survey

Dear Dr. Mungai,

Thank you for submitting your manuscript to PLOS Global Public Health. After careful consideration, we feel that it has merit but does not fully meet PLOS Global Public Health’s publication criteria as it currently stands. Therefore, we invite you to submit a revised version of the manuscript that addresses the points raised during the review process.

We look forward to receiving your revised manuscript.

Kind regards,

Marianne W. Mureithi, Ph.D

Academic Editor

Journal Requirements:

1. We see that your study includes live participants, but you have not included an Ethics Statement. Please update your manuscript file to include an Ethics Statement subsection to your Materials and Methods section. It should include:

i) The full name(s) of the Institutional Review Board(s) or Ethics Committee(s)

2. In the online submission form, you indicated that "The Kenya National Tuberculosis, Leprosy and Lung Disease program is the custodian of the 2016 Kenya Tuberculosis Prevalence Survey data.". All PLOS journals now require all data underlying the findings described in their manuscript to be freely available to other researchers, either 1. In a public repository, 2. Within the manuscript itself, or 3. Uploaded as supplementary information.

3. Please amend your detailed Financial Disclosure statement. This is published with the article, therefore should be completed in full sentences and contain the exact wording you wish to be published.

ii). State the initials, alongside each funding source, of each author to receive each grant.

iii). State what role the funders took in the study. If the funders had no role in your study, please state: “The funders had no role in study design, data collection and analysis, decision to publish, or preparation of the manuscript.”

Additional Editor Comments (if provided):

review your statistical model to estimate TB prevalence in participants with a very low underlying TB prevalence rate (those with no symptoms and a normal CXR). This has not been done with other publications of TB prevalence survey data and the authors have not provided any context/justification for their decision to follow this approach. Engage Someone with strong knowledge of modelling to guide through section.

Reviewers' comments:

Reviewer's Responses to Questions

**Comments to the Author**

1. Does this manuscript meet PLOS Global Public Health’s publication criteria? Is the manuscript technically sound, and do the data support the conclusions? The manuscript must describe methodologically and ethically rigorous research with conclusions that are appropriately drawn based on the data presented.

Reviewer #1: Partly

Reviewer #2: Yes

2. Has the statistical analysis been performed appropriately and rigorously?

Reviewer #1: I don't know

Reviewer #2: Yes

3. Have the authors made all data underlying the findings in their manuscript fully available (please refer to the Data Availability Statement at the start of the manuscript PDF file)?

Reviewer #1: Yes

Reviewer #2: No

4. Is the manuscript presented in an intelligible fashion and written in standard English?

Reviewer #1: Yes

Reviewer #2: Yes

5. Review Comments to the Author

Reviewer #1: The authors present an evaluation of CAD4TB v6 software using a very large and well-characterized data set. CAD software is ideally positioned for use in large-scale, roving surveys, so this is an ideal data set with which to conduct a CAD evaluation.

However, the added-value of the Bayesian modeling to estimate the TB burden in survey participants without TB symptoms or CXR signs suggestive of TB is unclear. The prevalence survey was designed to be the most accurate measure of TB prevalence possible and this type of modelling was not done to estimate disease burden in the prevalence survey publication you cited (#19). The background section does not introduce this approach or any studies using it to estimate disease burden. In this particular context, it seems like at best, this modelling approach would provide marginal value-add because people with no symptoms and a normal CXR have an extremely low chance of having TB, and that it may introduce falsely label participants as “cases” for the analysis. Given that this data set is among the highest-quality available for a CAD evaluation (e.g. scale, diagnostic algorithm, oversight for data quality, etc), the authors should provide more justification for using this modelling approach and the added value it provides for a CAD evaluation over focusing on the ~9,000 people with sputum test result data.

It is not clear from the manuscript text whether the CAD4TBv6 scores were as part of the Baysian modelling. The authors should provide more details in the manuscript body about what variables were used to build the model to estimate TB prevalence.

The authors should be more precise when writing about the approach. The analysis uses a modelled bacteriological reference standard. Sometimes the text reads as if the modelling does not exist. Example: Line 84 “…against the bacteriological reference standard used within the prevalence survey.”

The procedures for reading CXR images are not totally clear. The text on Line 108 indicates “either one of the Clinical Officers”. Does each image get two independent reads from two different Study Clinical Officers? Or were there two Study Clinical Officers per day, with one read by either Officer per image?

In the discussion about different software starting on Line 308, it should be noted that within each software new version are released on a regular basis. This analysis was done with a version of CAD4TB which is no longer commercially available. New users will receive CAD4TB v7.

Minor Comments:

Fig 1: The sum of groups under ‘Microbiological status assessed’ do not equal total (2,446 + 4,592 + 1,190 = 8,228). In the ‘Microbiologically-negative for TB’ box, fix the number formatting so it includes a comma.

I believe that references #6 and #16 are the same.

Reviewer #2: This study addresses a very important topic, the accuracy of computer aided chest X ray screening of the general population, within a national prevalence survey in Kenya. The article is well written but it does need to address several major and minor points.

Major points

Being a diagnostic accuracy study, the article should adhere to the STARD statement and include the checklist as an annex indicating the pages where each item is addressed.

Although I agree the results are promising, the discussion should better address limitations and shortcomings. It seems to me to be a bit too optimistic and a healthy scientific skepticism would not reduce the value of findings and CAD for community based screening, but provide a more rigorous and objective interpretation of results. For example, in line 352, “highly accurate” without mentioning the low specificity in older age groups and previous TB history (% of those groups should be provided too). Same in line 369-361. Are you sure your results support all these benefits? How can false positive bacteriological tests be reduced without also considering false positive CXR results? Also, a more exhaustive discussion on the role of screening and testing persons with and without symptoms should be provided. Finally, comparing the results with the 2021 TB screening recommendations from WHO and saying if this study results support the recommendations or if authors suggest modifications.

Specific comments:

Introduction

Line 65, I suggest to indicate the year of the WHO TB screening guidelines instead of “most recent”

Line 66, when authors say data come from “clinical settings” do they mean “and not from population based settings”? This could be specified.

Line 76, and specificity is probably lower than microbiological confirmation, is it not?

Line 87, “..vary between population groups..” could authors specify which population groups. For example something like “… vary between general population and high risk groups or symptomatic groups..”

Methods

Line 93, authors could specify if the Kenya National TB prevalence survey followed WHO standard methods for prevalence survey and include the reference

The study population section does not describe the study population. Were they only adults? Representative at national level (or province level)? HIV positive and negative? After reading the full article we learn about these items, but the population should be described here

Line 105 What is meant by “independently”? Two readers? Not clear. Where readers also blinded to clinical information (symptoms/no symptoms, HIV status, TB contact, etc. But symptoms would be the most important). This comes back in the discussion line 420.

Line 107 “..published definitions..” is insufficient for a study focused on X-ray reading. I suggest that you include a brief description of what was normal, abnormal. etc

Line 112 What if the person had no cough or could not produce sputum? Please clarify in the text

Line 114 please revise spelling of Löwenstein-Jensen

Line 115 How were discordances between Xpert, culture and smear managed? Please report the frequency of discordancies and the algorithm or decisions taken when this occurred

Line 117-119 revise spelling of Mycobacterium (if it requires capitalizing, italizing, etc)

Line 121 and symptoms?

What is meant by Sputum Xpert and/or culture positive? Do you mean either sputum or Xpert or culture positive?. Same in lines 132-133

Line 131 What is a “higher score”?

Line 143 this is the first time that authors mention if sputum was collected or not. It should be mentioned earlier as it is a key aspect of study design.

Line 146 please mention the primary study outcome earlier.

Why are predictive values not calculated, especially if the data comes from a population based study?

Line 148 The fact that cough more than two weeks was used as a criteria should be explained.

Lines 153-155 What is meant by “underlying status of active pulmonary TB” and by “underlying true TB status? Please clarify in the text

Line 155 Model diagnostics should be reported in the results section

Line 158 this is the first time authors mention the combined reference standard. Earlier in the text, authors indicate sputum Xpert or culture positive.

Line 160 what is cmdstanr interface.

Line 167 Which threshold?

Line 169 How can HIV status have missing data if testing was not performed? Please clarify.

Results

Line 183 Field reader is not used before in the text.

Figure 1 the flowchart provides new information that was not clear in the methods. Why is only a fraction of those assessed by CXR evaluated with microbiology?

Why is smear not reported, while it is mentioned at least twice in methods?

Please report data for narrative results presented in lines 197-199.

Line 202 The p<0.0001 is for each comparison?

Table 1. What about missing data? Is the p for age groups a p for trend? Please specify.

Line 280 first time chronic cough is mentioned. If you mean cough >2 weeks, please say so for consistency within the text.

Table 2, the title is not self explanatory and the subtitles are not clear. Credible intervals were not mentioned in the methods.

Discussion

Line 332. Xpert is highly specific in asymptomatic patients with low pre test probability?

Line 337 and 339, please indicate the sensitivity and specificity (in a rounded %).

Line 353, it is not clear how 50% of Xpert tests would be saved.

Line 363-364, this is not mentioned in methods (representative of the general pop and random selection). Same comment for line 397.

Line 401, what was anticipated?

Line 426. The fact that limitations are that results are dependent on the model assumptions is a generic statement. Which model assumptions do authors have doubts upon?

Line 430-431. This is not consistent with what is described in the methods. Extra pulmonary TB has not been mentioned at all earlier in the text.

Please include a discussion on why women were overrepresented in the survey.

In general, the impact of limitations on the study results should be further discussed.

The fact that HIV was not tested is a major limitation. Results could also be stratified by HIV positive by self report, as well as by symptom status among them (and considering 14 days of cough is used, instead of only 1 day).

6. PLOS authors have the option to publish the peer review history of their article (what does this mean?). If published, this will include your full peer review and any attached files.

**Do you want your identity to be public for this peer review?** For information about this choice, including consent withdrawal, please see our Privacy Policy.

Reviewer #1: No

Reviewer #2: **Yes: **Larissa Otero

---

## [Decision Letter · Decision Letter 1]

5 Jul 2022

PGPH-D-22-00030R1

Accuracy of computer-aided chest X-ray screening in the Kenya National Tuberculosis Prevalence Survey

Dear Mungai,

Thank you for submitting your manuscript to PLOS Global Public Health. After careful consideration, we feel that it has merit but does not fully meet PLOS Global Public Health’s publication criteria as it currently stands. Therefore, we invite you to submit a revised version of the manuscript that addresses the points raised during the review process.

We look forward to receiving your revised manuscript.

Kind regards,

Collins Otieno Asweto, PhD

Academic Editor

Journal Requirements:

Additional Editor Comments (if provided):

Reviewers' comments:

Reviewer's Responses to Questions

**Comments to the Author**

1. If the authors have adequately addressed your comments raised in a previous round of review and you feel that this manuscript is now acceptable for publication, you may indicate that here to bypass the “Comments to the Author” section, enter your conflict of interest statement in the “Confidential to Editor” section, and submit your "Accept" recommendation.

Reviewer #3: (No Response)

Reviewer #4: (No Response)

2. Does this manuscript meet PLOS Global Public Health’s publication criteria? Is the manuscript technically sound, and do the data support the conclusions? The manuscript must describe methodologically and ethically rigorous research with conclusions that are appropriately drawn based on the data presented.

Reviewer #3: Partly

Reviewer #4: Yes

3. Has the statistical analysis been performed appropriately and rigorously?

Reviewer #3: Yes

Reviewer #4: Yes

4. Have the authors made all data underlying the findings in their manuscript fully available (please refer to the Data Availability Statement at the start of the manuscript PDF file)?

Reviewer #3: Yes

Reviewer #4: (No Response)

5. Is the manuscript presented in an intelligible fashion and written in standard English?

Reviewer #3: No

Reviewer #4: Yes

6. Review Comments to the Author

Reviewer #3: The current title does not seem to add to the body of knowledge as you have already stated in the manuscript about the 2021 WHO recommendations on the use of CAD in place of human readers, in resource limited settings where radiologists are scarce, for interpreting digital chest X rays for screening and triage for TB disease among individuals aged 15 years and older in populations in which TB screening is recommended. However, the recommendation of “an adaptive approach to setting screening thresholds as it could further improve screening accuracy and efficiency” seems to be the key finding.

It would more beneficial if the title could be tweaked. The manuscript seems to be talking to something like this “Optimizing community-based TB screening accuracy and efficiency in high burden countries ( or resource limited settings): Lessons learned from the Kenya National TB Prevalence survey”

The methodology and result sections are not clear. They appear ambiguous and not well presented. There are a lot of findings but not well organized to have a good thought flow and there are a lot of repetition of same information in different ways.

Reviewer #4: The paper explores the use of CAD and CXRs on screening for active TB using secondary data from a large national TB survey.

The objectives of the study, and the methods including the statistical methods are clearly described and appropriate. Listed below are are few comments, that can be discussed further by the authors.

1. The aim of screening programmes is to pick up any suspected cases and do further investigations. In this case sensitivity becomes much more important than specificity because missing active cases of TB is more consequential and grave than misclassifying, negative individuals. The authors compared optimum and minimum TPP, within this range is there are 'desirable' cut off TPP that is clinically, economically etc preferable?

2. The authors did mention other important considerations before considering implementing the CAD system. The proposal or goal will be to see this being used middle to low income countries as well. The authors should consider in the discussion, what the main challenges of implementing this might be. It might be easy to get CXRs using portable machines, but is it easy to transmit/upload the images to a central place/server where these can the be read? Are there connectivity issues etc.

7. PLOS authors have the option to publish the peer review history of their article (what does this mean?). If published, this will include your full peer review and any attached files.

**Do you want your identity to be public for this peer review?** For information about this choice, including consent withdrawal, please see our Privacy Policy.

Reviewer #3: No

Reviewer #4: No

---

## [Decision Letter · Decision Letter 2]

19 Oct 2022

Accuracy of computer-aided chest X-ray in community-based tuberculosis screening: Lessons from the 2016 Kenya National Tuberculosis Prevalence Survey

PGPH-D-22-00030R2

Dear Dr Mungai,

We are pleased to inform you that your manuscript 'Accuracy of computer-aided chest X-ray in community-based tuberculosis screening: Lessons from the 2016 Kenya National Tuberculosis Prevalence Survey' has been provisionally accepted for publication in PLOS Global Public Health.

Best regards,

Associate Professor Suman Majumdar

Academic Editor

Thank you for your re-submission. As the reviewers have pointed out, kindly work with the editorial team to address the following in the final manuscript:

Line 278-281: Doublecheck p-value

Line 282: Doublecheck the p-values and be consistent with p<0.0001 and p<0.0003.

Reviewer Comments (if any, and for reference):

Reviewer's Responses to Questions

**Comments to the Author**

1. If the authors have adequately addressed your comments raised in a previous round of review and you feel that this manuscript is now acceptable for publication, you may indicate that here to bypass the “Comments to the Author” section, enter your conflict of interest statement in the “Confidential to Editor” section, and submit your "Accept" recommendation.

Reviewer #4: All comments have been addressed

Reviewer #5: All comments have been addressed

2. Does this manuscript meet PLOS Global Public Health’s publication criteria? Is the manuscript technically sound, and do the data support the conclusions? The manuscript must describe methodologically and ethically rigorous research with conclusions that are appropriately drawn based on the data presented.

Reviewer #4: Yes

Reviewer #5: Yes

3. Has the statistical analysis been performed appropriately and rigorously?

Reviewer #4: Yes

Reviewer #5: I don't know

4. Have the authors made all data underlying the findings in their manuscript fully available (please refer to the Data Availability Statement at the start of the manuscript PDF file)?

Reviewer #4: No

Reviewer #5: Yes

5. Is the manuscript presented in an intelligible fashion and written in standard English?

Reviewer #4: Yes

Reviewer #5: Yes

6. Review Comments to the Author

Reviewer #4: (No Response)

Reviewer #5: The authors have adequately responded to the comments raised and the manuscript is almost there. I only have minor comments/clarifications

Line 77-80: Please add a quick discussion line in the background regarding the different versions of CAD4TB considering that there have been previous versions and you used v6 and you discussed availability of v7.

Line 278-281: Doublecheck p-value

Line 282: Doublecheck the p-values and be consistent with p<0.0001 and p<0.0003.

7. PLOS authors have the option to publish the peer review history of their article (what does this mean?). If published, this will include your full peer review and any attached files.

**Do you want your identity to be public for this peer review?** For information about this choice, including consent withdrawal, please see our Privacy Policy.

Reviewer #4: No

Reviewer #5: **Yes: **Tafireyi Marukutira
